# Characterizing Bodyweight-Supported Treadmill Walking on Land and Underwater Using Foot-Worn Inertial Measurement Units and Machine Learning for Gait Event Detection

**DOI:** 10.3390/s23187945

**Published:** 2023-09-17

**Authors:** Seongmi Song, Nathaniel J. Fernandes, Andrew D. Nordin

**Affiliations:** 1Division of Kinesiology, Texas A&M University, College Station, TX 77843, USA; songseongmi@tamu.edu; 2Department of Computer Science & Engineering, Texas A&M University, College Station, TX 77843, USA; njfernandes24@tamu.edu; 3Department of Biomedical Engineering, Texas A&M University, College Station, TX 77843, USA; 4Texas A&M Institute for Neuroscience, Texas A&M University, College Station, TX 77843, USA

**Keywords:** machine learning, gait event detection, reduced gravity, mechanical body weight support, underwater walking

## Abstract

Gait rehabilitation commonly relies on bodyweight unloading mechanisms, such as overhead mechanical support and underwater buoyancy. Lightweight and wireless inertial measurement unit (IMU) sensors provide a cost-effective tool for quantifying body segment motions without the need for video recordings or ground reaction force measures. Identifying the instant when the foot contacts and leaves the ground from IMU data can be challenging, often requiring scrupulous parameter selection and researcher supervision. We aimed to assess the use of machine learning methods for gait event detection based on features from foot segment rotational velocity using foot-worn IMU sensors during bodyweight-supported treadmill walking on land and underwater. Twelve healthy subjects completed on-land treadmill walking with overhead mechanical bodyweight support, and three subjects completed underwater treadmill walking. We placed IMU sensors on the foot and recorded motion capture and ground reaction force data on land and recorded IMU sensor data from wireless foot pressure insoles underwater. To detect gait events based on IMU data features, we used random forest machine learning classification. We achieved high gait event detection accuracy (95–96%) during on-land bodyweight-supported treadmill walking across a range of gait speeds and bodyweight support levels. Due to biomechanical changes during underwater treadmill walking compared to on land, accurate underwater gait event detection required specific underwater training data. Using single-axis IMU data and machine learning classification, we were able to effectively identify gait events during bodyweight-supported treadmill walking on land and underwater. Robust and automated gait event detection methods can enable advances in gait rehabilitation.

## 1. Introduction

Gait analysis plays a crucial role in evaluating locomotor function and diagnosing movement disorders [1,2]. Assessing biomechanical and neural signals during locomotion requires the identification of gait events during each step or stride of the gait cycle, including the instant the foot contacts and leaves the ground (initial contact and foot off, respectively). Traditional gait event detection approaches rely on ground reaction force measures and video-based motion capture, but the associated costs and spatial requirements are often impractical for continuous real-world gait monitoring [3]. Ground reaction force measures require a stable mounting surface and direct contact with the force plate [4], typically only available in controlled laboratory settings. Video-based motion capture relies on complex camera configurations that may still suffer from marker occlusion and gait event detection errors [4]. Alternatively, pressure sensors or footswitches can be portably inserted into footwear but may lack durability, sufficient sampling rate, and/or battery life.

Lightweight and wireless inertial measurement unit (IMU) sensors incorporate six-degree-of-freedom accelerometer and gyroscope recordings and can be used for gait event detection without the need for force plate or video recordings by analyzing the acceleration and angular velocity patterns of body segments during locomotion [5,6,7]. Initial foot-ground contact events have been identified based on local peaks in the vertical acceleration versus time profile of IMU sensors placed on the foot during walking [8,9], in addition to the instant when the vertical acceleration of the foot crosses zero from negative to positive [8,10,11]. Foot-off events have similarly been identified based on local peaks in the angular velocity versus time profile of IMU sensors placed on the shank segment during walking [12], and using the instant when the angular velocity of the foot crosses zero from positive to negative in the sagittal plane, around the mediolateral axis [13].

Gait event detection often requires the identification of hundreds or thousands of initial contact and foot-off events during continuous locomotion based on user-defined criteria from biomechanical signals. Step-to-step variation and gait variability among individuals can make robust gait event detection challenging and time-consuming when inspecting event tagging accuracy. A possible solution to overcome these challenges is to rely on machine learning algorithms that can be trained on event-labeled ground truth IMU data to detect gait events in another dataset. This method can improve accuracy and automate event detection procedures [14,15] but is dependent on sensor placement, orientation, and the selected IMU signal component [16,17]. The placement of an IMU sensor under the arch of the foot, within the sole of the shoe, achieved better gait event detection accuracy compared to other locations on the lower limb during running [18] and walking [19]. Independent of sensor location, rotational velocity around the mediolateral axis and acceleration along the anterior-posterior axis have demonstrated robust gait event detection accuracy during walking [16].

During gait training and rehabilitation, bodyweight unloading can improve walking function for individuals with mobility impairments, reducing balance demands and impacts on the lower extremities [20,21]. Bodyweight support is provided by applying upward forces to the body, conventionally using overhead mechanical harness systems. With the advent of smart rail-type [22] or robotic bodyweight support systems [23,24,25], individuals are provided the option to walk overground or on a treadmill. Similar to mechanical bodyweight support mechanisms, water immersion provides bodyweight unloading due to the buoyancy of the human body, while also introducing greater drag forces due to water viscosity, which are beneficial for gait rehabilitation [26,27,28] and exercise [29]. Despite the advantages of underwater gait rehabilitation, quantifying underwater gait has remained challenging due to the reliance on costly submerged force plates [30,31] and camera-based motion capture [32,33]. Using video recording methods, Volpe et al. [32] identified altered spatiotemporal gait parameters and sagittal plane lower limb kinematics among individuals with Parkinson’s disease who underwent aquatic therapy. Compared to camera-based motion capture, the accuracy of underwater knee joint angle measurements has been demonstrated [34], but few studies have analyzed underwater gait using IMU sensors [34,35,36]. Automated gait event detection using IMU sensor data provides a critical opportunity for facilitating real-world human locomotion studies.

This study aimed to assess the use of machine learning algorithms to identify gait events based on features from foot segment rotational velocity around the mediolateral axis using foot-worn IMU sensors during bodyweight-supported treadmill walking on land and underwater.

We hypothesized that (1) foot segment rotational velocity features would enable accurate gait event detection during bodyweight-supported treadmill locomotion on land. We further hypothesized that (2) robust gait event detection on land would enable accurate gait event detection during underwater bodyweight-supported treadmill locomotion.

## 2. Materials and Methods

### 2.1. Study Protocol

#### 2.1.1. Subject Information

Twelve healthy subjects (5 females and 7 males, age: 23.2 years, height: 171.9 cm, mass: 71.8 kg) completed on-land treadmill walking with overhead mechanical bodyweight support (Figure 1A). Three subjects (1 female and 2 males, age: 24 years, height: 170.8 cm, mass: 67.9 kg) completed underwater bodyweight-supported treadmill walking, gathering IMU sensor data alongside reliable ground truth measurements (Figure 1B). Subjects had no previous or existing lower extremity injuries or neuropathies.

#### 2.1.2. On-Land Body-Weight-Supported Treadmill Walking

Subjects walked on the treadmill (M-Gait, Motek, Amsterdam, The Netherlands) at four different walking speeds (0.4 m/s, 0.8 m/s, 1.2 m/s, and 1.6 m/s) and three different bodyweight support conditions (no BWS, 30% BW, 50% BW). Our goal was to include slow and fast gait speeds, spanning paces that are common among individuals with gait deficits, the typical preferred walking speed of able-bodied individuals (1.2 m/s), and comfortable fast walking, prior to the preferred walk-to-run transition speed [37]. Bodyweight support (BWS) levels were selected based on relationships between water submersion depth relative to specific anatomical landmarks. Submerging the human body into water up to the standing height of the superior iliac crests of the pelvis achieves approximately 30% BWS, and the xiphoid process on the sternum achieves approximately 50% BWS [38]. During on-land BWS treadmill walking, we calibrated the mechanical harness system to provide 30% and 50% BWS to match underwater testing. The gait speed order was randomized, and subjects walked on the treadmill at the same speed without BWS between each randomized BWS condition. The treadmill walking duration was 4 min in each condition.

We instrumented subjects with two IMU sensors (sampling rate: 2000 Hz, Wave, Cometa, Italy), one on top of each foot (between talus and center of metatarsal bones) to measure accelerations and angular velocities of the foot segment during treadmill walking (Figure 1A). Ground reaction force data were measured from a split-belt treadmill (sampling rate: 2000 Hz). We also collected 64 channels of electroencephalography (EEG), 16 channels of surface electromyography (EMG), and motion capture data, but we did not include these data in this paper. All data were time-synchronized (Nexus 2.14, Vicon, Oxford, UK).

#### 2.1.3. Underwater Bodyweight-Supported Treadmill Walking

Subjects walked on the underwater treadmill (HYDROWORX 1200, Middletown, PA, USA) at four speeds: 0.4 m/s, 0.8 m/s, 1.2 m/s, and 1.6 m/s. Water immersion depths corresponded to the superior iliac crests of the pelvis (waist-level, ~30% BWS; [38]) and the xiphoid process on the sternum (chest-level, ~50% BWS; [38]) for each subject. The treadmill walking duration was 3 min in each condition.

During testing, subjects wore socks over waterproofed wireless foot pressure insoles that contained embedded IMU sensors (sampling rate: 100 Hz, OpenGo Sensor, Motion, Munich, Germany; Figure 1E). Data from the same two IMU sensors that were placed on top of each foot during on-land testing were also recorded, but because of precise temporal synchronization of the foot pressure sensors with the embedded IMU, and parallel orientation of the mediolateral axes between each IMU sensor, we used the IMU sensor embedded within the pressure sensing insoles for analysis.

### 2.2. Data Analysis 

#### 2.2.1. On-Land Body-Weight-Supported Treadmill Walking Data

We generated custom MATLAB scripts for data processing (version: 2021b, MathWorks Inc., Natick, MA, USA). During analysis, we extracted a central 3 min portion of the 4 min treadmill walking condition. Ground truth gait events were identified from ground reaction force data after downsampling the vertical ground reaction force and IMU data to 100 Hz and low pass filtering (4 Hz cutoff). We defined initial contact (IC) and foot-off (FO) events based on the instant the vertical ground reaction force exceeded and fell below 10 N [39,40]. The IC class was labeled as 1, the FO class was labeled as 2, and other than IC and FO was labeled as 0. To address the class imbalance issue of initial contact (IC)and foot-off (FO) events, as well as events occurring within small time intervals, we employed data augmentation techniques to re-label the IC and FO classes from individual data points to boundary events. Specifically, we expanded the classes from a single data point to ten data points centered around the original data point. Furthermore, in our study, we employed machine learning algorithms to define the stance phase and the swing phase. Nonetheless, our findings revealed that utilizing a data augmentation approach yielded higher accuracy compared to the approach of categorizing the phases into stance and swing (refer to Appendix A).

The resampled and low pass filtered IMU data from each sensor were amplitude normalized for each subject based on the maximum value during treadmill walking without BWS in the fastest walking speed condition (1.6 m/s).

#### 2.2.2. Underwater Bodyweight-Supported Treadmill Walking Data

We analyzed the data using custom MATLAB scripts (version: 2021b, MathWorks Inc., Natick, MA, USA). During analysis, we extracted a central around 2 min portion of the 3 min treadmill walking condition. Ground truth gait events for the classification algorithm were identified from the wireless foot pressure sensor data. We calculated the total force from the foot pressure sensors and low pass filtered (4 Hz cutoff) the foot pressure and IMU data. We defined initial contact (IC) and foot-off (FO) events based on the instant the vertical ground reaction force exceeded and fell below 10 N [41]. Data augmentation procedures were identical to those used for mechanical BWS walking data in Section 2.2.1. 

The low pass filtered IMU data from each sensor were amplitude normalized for each subject based on maximum value during underwater treadmill walking at waist level in the fastest walking speed condition (1.6 m/s).

### 2.3. Features and Classifications

We used the Python Scikit-learn library [42] to perform event classification. During preliminary analyses, we applied random forest, decision tree, support vector machine, and k-nearest neighbor machine learning algorithms. Because the random forest approach returned the greatest accuracy and testing speed (refer to Appendix B), we used this approach during the subsequent analyses (number of trees in the forest = 100, maximum tree depth = none).

Informed by studies prior to examining on-land and underwater BWS walking, we focused our analysis on the sagittal plane foot segment angular velocity from the IMU gyroscope data around the mediolateral axis (Y-axis, Figure 1D). Feature extraction relied on the low pass filtered gyroscope data, along with statistical metrics including root mean square, skewness, kurtosis, and variance, computed with a window size of five data points. We also included derivative values and zero-crossing information, where a zero-crossing was assigned as 1, and other than zero-crossing points were assigned as 0. 

#### 2.3.1. Gait Event Detection during Bodyweight-Supported Treadmill Walking on Land

To test our first hypothesis, we first compared the similarity of the sagittal plan foot segment angular velocity time series data between treadmill walking at full bodyweight (no BWS) and during BWS walking on land. To estimate gait events during BWS walking, we used training data from all subjects during treadmill walking on land without BWS. We allocated 80% of the treadmill walking data without BWS for training the model and predicting gait events for 100% of the BWS walking data. Through a process of random shuffling, we reorganized the gait cycles, sampled among subjects and study conditions, and designated 80% of the dataset for training purposes, and the remaining 20% as the test set. We also trained the model with data from 10 subjects in the BWS walking dataset and evaluated its performance using the remaining 2 subjects from the same dataset. 

#### 2.3.2. Gait Event Detection during Underwater Bodyweight-Supported Treadmill Walking

To test our second hypothesis, we compared the similarity of the foot segment angular velocity time series data between BWS treadmill walking on land and during BWS treadmill walking underwater. To estimate gait events during underwater BWS walking, we used a model trained on data during BWS walking on land. We allocated 80% of the data during BWS walking on land for training the model and predicting gait events for 100% of the underwater BWS walking data. Once again, we randomly shuffled a selection of gait cycles sampled among subjects and study conditions and designated 80% of the dataset for training purposes, and the remaining 20% as the test set. Finally, we estimated gait events during underwater BWS walking from a model trained specifically on underwater BWS walking data, allocating data from two subjects for training and using the remaining data from one subject for testing. 

#### 2.3.3. Classification Evaluation

We assessed machine learning classification performance using a confusion matrix [42] to evaluate *accuracy* (Equation (1); [42,43]), *precision* (Equation (2); [42]), *recall* (Equation (3); [42]), and *F1 score* (Equation (4); [42]), and employed 10-fold cross-validation [42], dividing the dataset into 10 folds, training the model on 9 folds, and testing the remaining fold.
(1)$$Accuracy=\frac{True\;Positives+True\;Negatives}{True\;Positives+True\;Negatives+False\;Positive+False\;Negative}$$
(2)$$Precision=\frac{True\;Positives}{True\;Positives+False\;Positives}$$
(3)$$Recall=\frac{True\;Positives}{True\;Positives+False\;Negatives}$$
(4)$$F1\;Score=2*\;\frac{Precision*\;Recall}{Precision+Recall}$$

### 2.4. Data Similarity Analysis

We compared sagittal plane foot segment angular velocity versus time profiles using cross-correlation coefficients between paired BWS and gait speed conditions. Five strides were extracted from the middle of the dataset and resampled for comparison. Resampling rates differed by walking speed due to contrasting stride rates relative to the IMU sampling rate (0.4 m/s: 800 samples; 0.8 m/s and 1.2 m/s: 600 samples; and 1.6 m/s: 400 samples). We iteratively repeated the process ten times, totaling 50 steps, and calculated the average correlation coefficient values for each pairwise condition comparison using the “*corrcoef*” function in MATLAB.

## 3. Results

### 3.1. Sagittal Plane Foot Segment Angular Velocity Characteristics

Foot segment angular velocity versus time profiles around the mediolateral axis varied among BWS and gait speed conditions (Figure 2 and Table 1). Slower gait speeds showed lesser cross-correlation coefficients compared to faster gait speeds when evaluating pairwise comparisons between BWS conditions (Table 1). Pairwise comparisons between on-land and underwater BWS treadmill walking conditions also showed lesser cross-correlation coefficients.

### 3.2. Gait Event Detection during Bodyweight-Supported Treadmill Walking on Land

#### 3.2.1. Training (80% Data): No BWS, Test (100% Data): BWS on Land (Table 2 and Figure 3A)

Using the random forest algorithm for identifying gait events from IMU sagittal plane foot segment angular velocity data, we achieved 95% detection accuracy using training data during treadmill walking without BWS to predict gait events during BWS treadmill walking on land. Precision and recall rates were 86% and 85%, respectively, for detecting initial contact events, and 87% and 85%, respectively, for foot-off detection.

**Table 2 sensors-23-07945-t002:** Gait Event Estimation Results Using Random Forest Classification.

Training Data	Test Data	Gait Events	Precision	Recall	F1-Score	Accuracy (10-Fold Cross Validation)
No BWS on-land 80%	BWS on-land 100%	NaN	0.97	0.98	0.97	0.95
		IC	0.86	0.85	0.86	
		FO	0.87	0.85	0.86	
BWS on-land (10 subjects)	BWS on-land (2 subjects)	NaN	0.97	0.97	0.97	0.95
		IC	0.79	0.82	0.80	
		FO	0.88	0.86	0.87	
BWS on-land 80%	Underwater 100%	NaN	0.90	0.75	0.82	0.71
		IC	0.07	0.16	0.10	
		FO	0.30	0.62	0.40	
Underwater (2 subjects)	Underwater (1 subject)	NaN	0.94	0.95	0.94	0.89
		IC	0.49	0.41	0.44	
		FO	0.66	0.61	0.64	

**Figure 3 sensors-23-07945-f003:**
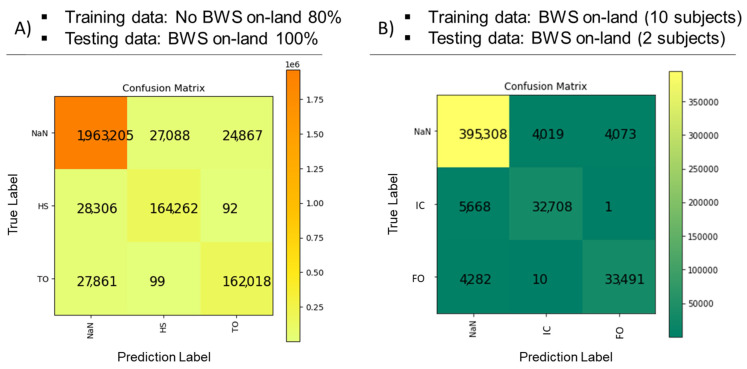
Confusion matrices show model classification performance. (**A**) Gait event prediction during bodyweight-supported (BWS) treadmill walking on land using a model trained on 80% of the dataset during treadmill walking on land without BWS (no BWS). (**B**) Gait event prediction during BWS walking on land using a model trained on 10 subjects’ datasets during BWS treadmill walking on land. IC is initial contact events, and FO is foot-off events. The x-axis shows the prediction labels and the y-axis shows the true labels.

#### 3.2.2. Training (Ten Subjects’ Data): BWS on Land, Test (Two Subjects’ Data): BWS on Land (Table 2 and Figure 3B)

We achieved 95% accuracy for detecting gait events from the IMU sagittal plane foot segment angular velocity data when trained and tested on BWS treadmill walking data on land. Precision and recall rates were 79% and 82%, respectively, for detecting initial contact events, and 88% and 86% for foot-off detection.

### 3.3. Gait Event Detection during Underwater Bodyweight-Supported Treadmill Walking

#### 3.3.1. Training (80% Data): BWS on Land, Test (100% Data): Underwater BWS (Table 2 and Figure 4A)

Using the random forest algorithm for identifying gait events from IMU sagittal plane foot segment angular velocity data, we achieved 71% detection accuracy using training data during BWS treadmill walking on land to predict gait events during BWS treadmill walking underwater. Precision and recall rates were 7% and 16%, respectively, for detecting initial contact events, and 30% and 62%, respectively, for foot-off detection.

**Figure 4 sensors-23-07945-f004:**
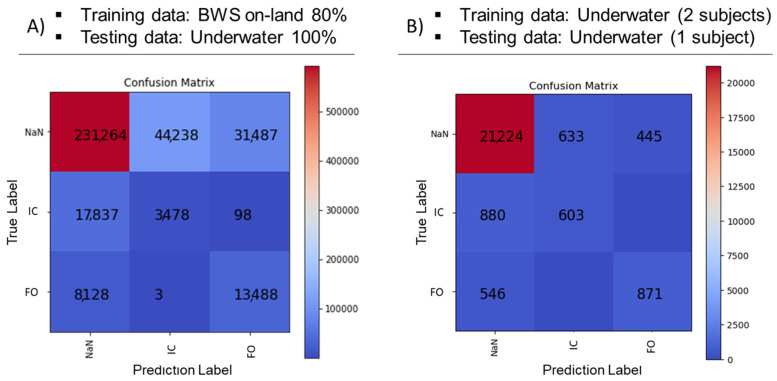
Confusion matrices show model classification performance. (**A**) Gait event prediction during underwater bodyweight-supported (BWS) treadmill walking using a model trained on 80% of the dataset during BWS treadmill walking on land. (**B**) Classification performance for predicting gait events during underwater BWS walking using a model trained on data from two subjects during underwater BWS treadmill walking. IC is initial contact events, and FO is foot-off events. The x-axis shows the prediction labels, and the y-axis shows the true labels.

#### 3.3.2. Training (Two Subjects’ Data): Underwater BWS and Test Data (One Subject’s Data): Underwater BWS (Table 2 and Figure 4B)

We achieved 89% accuracy for detecting gait events from the IMU sagittal plane foot segment angular velocity data when trained and tested on underwater BWS treadmill walking data. Precision and recall rates were 49% and 41%, respectively, for detecting initial contact events, and 66% and 61%, respectively, for foot-off detection.

## 4. Discussion

Our aim was to assess the use of machine learning algorithms to identify gait events based on sagittal plane foot segment rotational velocity features using foot-worn IMU sensors during bodyweight-supported treadmill walking on land and underwater. In partial agreement with our hypotheses, (1) the random forest machine learning approach enabled accurate gait event detection based on sagittal plane foot segment rotational velocity features during bodyweight-supported treadmill walking on land. However, (2) gait event detection during bodyweight-supported treadmill walking underwater was poorly predicted when the same random forest algorithm was trained on data recorded during on-land bodyweight-supported treadmill walking.

### 4.1. Contrasting Sagittal Plane Foot Segment Angular Velocity during Bodyweight-Supported Treadmill Walking on Land and Underwater

We identified similarities between treadmill walking on land with and without BWS based on foot segment angular velocity around the mediolateral axis, but these similarities failed to translate to BWS treadmill walking underwater. Cross-correlation analysis showed the greatest similarity between treadmill walking on land without BWS and 30% BWS on land, while walking underwater at chest level was least similar to walking on land among walking speeds (Table 1). We anticipated greater similarity between BWS treadmill walking on land and underwater based on the assistive upward forces applied to the body opposing gravity, however, ground truth initial contact events from force plate and foot pressure sensors showed timing discrepancies when compared to the sagittal foot segment plane angular velocity profile. Specifically, during BWS treadmill walking on land, initial contact events coincided with zero-crossing points in the sagittal foot segment angular velocity versus time profile, but initial contact events occurred prior to the zero-crossing points during underwater walking (Figure 2). Initial contact timing discrepancies could be attributed to the external fluid forces present underwater that are absent on land [26], altering foot segment plantar and dorsiflexion velocity, which should be taken into consideration when estimating gait events in underwater environments using IMU sensors.

### 4.2. Gait Event Detection during Bodyweight-Supported Treadmill Walking

We achieved a high level of accuracy when estimating gait events during BWS walking on land when training the model using data from treadmill walking on land without BWS (95% accuracy) and with BWS (95% accuracy) (Table 2). High classification accuracy can be attributed to the similarities between sagittal plan foot segment velocity profiles and event timings identified during cross-correlation analysis (Table 1). Using the random forest algorithm for gait event detection on land provides an opportunity for automated gait event detection during BWS treadmill walking on land relying on sagittal plane foot segment angular velocity features alone. Previous studies have obtained an accuracy of 95–98% with a combination of accelerometer and gyroscope IMU data using machine learning algorithms [14,15,44]. Nevertheless, there are notable benefits associated with utilizing single-axis data. By relying solely on data from a single axis, the sensor location on the foot can be more generalized for gait event detection. Previous studies have shown that gait event detection accuracy can vary based on the sensor’s locations [18,19]. The foot commonly accommodates IMU sensor placements on the top and under the foot, which can vary based on specific purposes and design considerations. Our approach enables the effective application of the algorithm to both sensor locations, regardless of their placement on the foot. Moreover, this approach simplifies the data processing and feature extraction steps of the machine learning algorithm. As a result, it reduces the complexity of the model and minimizes the computational requirements involved in the gait event detection process.

Although gait event detection accuracy was unsatisfactory when using sagittal foot segment angular velocity features to predict gait events during underwater walking from a model trained on BWS treadmill walking on land (71% accuracy), we achieved a better accuracy when detecting gait events using a model trained on underwater treadmill walking data (89%). Based on the observed cross-correlation discrepancies between BWS on land compared to underwater treadmill walking, changes in sagittal plane foot segment angular velocity timing required model training specific to underwater treadmill gait. Consistent with our own findings, previous studies have highlighted the necessity of adapting models to account for the unique biomechanical characteristics of underwater walking. Kinematic investigations have revealed that underwater walking and on-land walking share similar modulation patterns [35,45,46]. However, individuals finely adjust their movements to meet the specific biomechanical requirements of walking in underwater environments. In terms of sagittal plane kinematics, studies demonstrated comparable joint angle patterns while walking in water, although there are variations in flexion and extension angles, with greater knee and hip flexion and increased ankle dorsiflexion. [35,45]. Additionally, spatiotemporal gait parameters slightly vary underwater, characterized by longer stride duration and shorter stride distance [35]. Furthermore, calf muscle activation decreases during the stance phase, while the activations of the rectus femoris and biceps femoris muscles increase during the swing phase [47,48]. In addition, Fantozzi et al. [46]. observed distinct gait patterns in water among both healthy elderly individuals and young adults, which were influenced by age, walking speeds, and the interaction between these factors. The results combined with previous research emphasized the significance of gaining a comprehensive understanding of underwater walking.

### 4.3. Limitations and Future Directions

Our analysis relied on a relatively small number of healthy young adults, which may limit the generalizability of our results. To overcome the limited number of participants in our study, we recorded continuous walking data during four distinct gait speeds and three varied levels of bodyweight support. Our dataset therefore consisted of approximately 246,000 data points (on land) and 144,000 data points (underwater) per subject, derived from 180-s and 120-s recording intervals in each respective condition. The possibility of underrepresented between-subject variability was compensated by the number of data points and within-subject variability among gait speed and bodyweight support conditions. Gait event classification performance was robust to step-to-step variations throughout the dataset caused by altered foot segment angular velocity versus time profiles. Because clinical populations often present slower walking speeds and altered gait biomechanics that can affect foot-ground interactions, accurate gait event detection among our varied experimental conditions is promising for gait rehabilitation applications where participant numbers are often limited.

Despite using only sagittal plane foot segment angular velocity data for identifying gait events during BWS treadmill walking on land and underwater, we incorporated additional statistical features from this signal to enhance classification accuracy. We believe that our simple approach of relying exclusively on foot segment rotational velocity around the mediolateral axis provides a biomechanically sound measure related to gait event timing that is readily available from IMU sensor data. Nevertheless, specialized models and algorithms for gait event estimation during underwater walking may improve when incorporating other potential IMU signal features in different walking conditions.

## 5. Conclusions

We were able to detect gait events during BWS treadmill walking on land and underwater based on sagittal plane foot segment angular velocity data using random forest machine learning classification. Although dependent on biomechanical changes during underwater compared to on-land treadmill walking, we achieved high accuracy, precision, and recall rates for gait event estimation across a range of gait speeds and BWS conditions. Because gait event detection is necessary for analyzing biomechanical and neural data for studying human locomotion, robust gait event detection methods based on IMU sensor recordings provide opportunities to conduct studies in environments that were previously challenging or not possible.

## Figures and Tables

**Figure 1 sensors-23-07945-f001:**
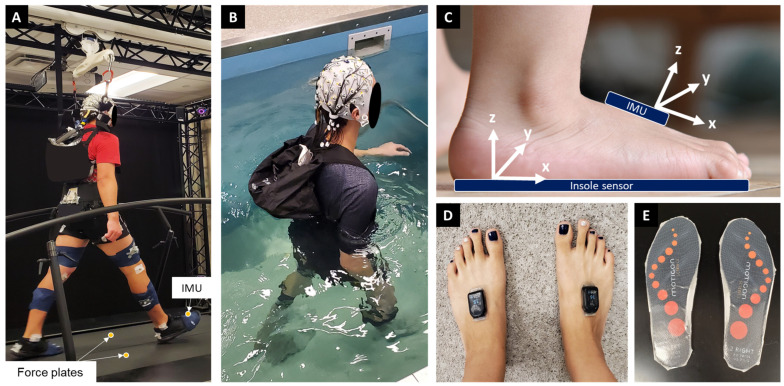
Experimental setup. (**A**) Subject walking on the force measuring treadmill on land with overhead mechanical bodyweight support. (**B**) Subject walking on the underwater treadmill. (**C**) Sagittal view of the foot, showing the IMU sensor, insole sensor placement, and axis orientations. The IMU sensor on top of the foot was positioned between the talus and the center of the metatarsal bones. The foot pressure insole was situated beneath the foot and contained an embedded IMU sensor. (**D**) Overhead transverse plane view of IMU foot sensor placement. (**E**) Water-proofed MOTICON pressure insole sensors.

**Figure 2 sensors-23-07945-f002:**
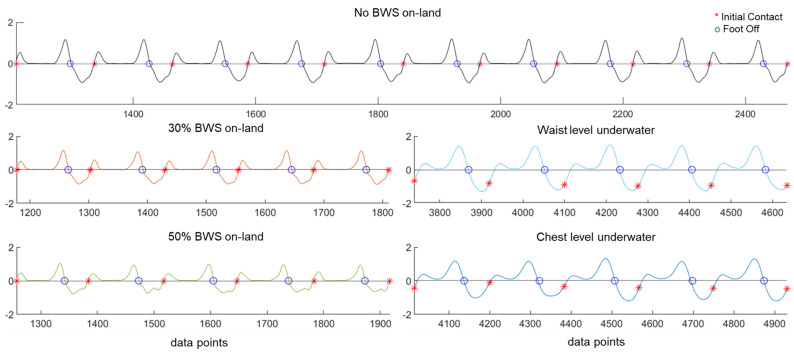
Sagittal plane foot segment angular velocity versus time during bodyweight-supported (BWS) treadmill walking conditions (1.2 m/s walking). On-land treadmill walking at full bodyweight (no BWS: top, black line), 30% BWS (left, middle row, orange line); 50% BWS (left, bottom row, green line). Underwater treadmill walking at waist level depth (right, middle row, light blue line) and underwater walking at chest level water depth (right, bottom row, dark blue). Foot segment angular velocity values were amplitude normalized to the maximum value in the fastest walking speed condition (1.6 m/s). Initial contact (red *), and foot-off events (blue circles) are shown in each condition.

**Table 1 sensors-23-07945-t001:** Mean Sagittal Plane Foot Segment Angular Velocity Versus Time Pairwise Cross-Correlations Between Bodyweight Support (BWS) Conditions at Each Gait Speed.

Comparison Pairs	Correlation Coefficient				
	0.4 m/s	0.8 m/s	1.2 m/s	1.6 m/s	Average
No BWS on-land—30% BWS on-land	0.81	0.96	0.99	0.97	0.93
No BWS on-land—50% BWS on-land	0.69	0.85	0.96	0.93	0.86
No BWS on-land—Waist level underwater	0.68	0.78	0.67	0.66	0.70
No BWS on-land—Chest level underwater	0.50	0.71	0.69	0.66	0.64
30% BWS on-land—Waist level underwater	0.72	0.77	0.67	0.67	0.71
50% BWS on-land—Chest level underwater	0.51	0.79	0.69	0.71	0.67
30% BWS on-land—50% BWS on-land	0.76	0.90	0.96	0.96	0.90
Waist level underwater—Chest level underwater	0.72	0.86	0.86	0.96	0.85
Average	0.67	0.83	0.81	0.81	

## Data Availability

The data presented in this study are available on request from the corresponding author.

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
