# Peer review of "Characterizing Bodyweight-Supported Treadmill Walking on Land and Underwater Using Foot-Worn Inertial Measurement Units and Machine Learning for Gait Event Detection"

_sensors, 2023, doi:10.3390/s23187945_

Round 1

Reviewer 1 Report

I appreciate the opportunity to review this manuscript entitled, “Characterizing Bodyweight-Supported Treadmill Walking On-Land and Underwater Using Foot-Worn Inertial Measurement Units and Machine Learning for Gait Event Detection.” Accurately detecting gait events, particularly initial contact and foot-off, is important because many spatiotemporal gait parameters are dependent on detecting those events. The study’s purpose is well justified, and the paper is generally well written. Summarizing, the authors report the IMUs used in the study accurately detected the gait events during on-land BWS treadmill walking, but that accurate gait event detection in the underwater treadmill condition required additional machine learning with random forest classification.

My following specific comments are intended to help the authors improve the manuscript and are directed toward the following questions:

Does the introduction provide sufficient background and include all relevant references?
1. Both the Abstract (line 13) and the Introduction narrative (line 39) state that traditional methods for gait event detection face limitations in real-world settings. Describe or provide examples of those limitations. Identifying the limitations and describing how your approach may ameliorate those limitations would provide better justification for your study.

2. As a general comment, describing the key gait events of “heel contact” and “toe off” reflects old terminology in gait research. More contemporary descriptions of these events are labeled “initial contact” and “foot off” because not all individuals make initial contact at the heel and, likewise, not all individuals have last contact at the toes. I recommend you use the more contemporary descriptors of “initial contact” and “foot off” throughout your manuscript.

Are all the cited references relevant to the research?
3. Yes, references are appropriate and relevant.

Is the research design appropriate? 
4. My primary critiques regarding the study design relate to sample size. First, there is no power analysis/sample size justification provided. Please describe how 12 subjects were determined a priori to be sufficient for your purpose. Second, only 3 subjects participated in the underwater walking condition (line 93). This is the primary limitation in the study. Can you justify that a sample of 3 subjects is sufficient to quantify accuracy of gait event detection? I do not claim expertise in use of machine learning algorithms, though I have used some forms of machine learning in our own work (e.g., Classification and Regression Tree [CART] models, Chi-Square Automatic Interaction Detection [CHAID] models) and I have been under the impression that machine learning algorithms generally require large training samples (for example, Figueroa RL, et al. Predicting sample size required for classification performance. BMC Medical Informatics and Decision Making. 2012;12:8).

Are the methods adequately described?
5. Please justify the standardized speed selections and %BW selections used in the study (lines 99-100).

6. EEG and EMG data are not reported in the manuscript. I recommend you omit your description of EEG and EMG data collection from the Methods section of the paper (lines 107-108). It is unnecessary information, given your paper’s purpose.

7. At lines 162-166, the authors briefly describe the use of four machine learning algorithms included random forest, decision tree, support vector machine, and k-nearest neighbor algorithms. Then they state the random forest approach was most accurate, and therefore selected for ongoing analysis. But there’s no data provided to support that assertion. How can the reader trust that the random forest algorithm provided the greatest accuracy without seeing the data? Or providing a reference to a study supporting the assertion? Should you consider publishing that analysis, perhaps as a technical report?

8. Can you cite references supporting the operational definitions and equations used for determining accuracy, precision, recall, and a F1 score (lines 193-202)?

9. I typically expect a Methods section to conclude with a Data Analysis/Statistical Analysis subsection. That is missing from the manuscript, though some of the analysis approach is described in the Classification Analysis subsection in which accuracy, precision, recall, and F1 score are introduced (per comment above). But other analyses were used, including confusion matrices (Figs. 3 & 4 in the Results section). Since that is a method used to analyze the data, it should be described in the Methods section.

Are the results clearly presented?
10. The narrative description of results is clearly written and the associated tables/figures support the narrative.

Are the conclusions supported by the results? 
11. The Discussion section is well-written and the study’s findings are appropriately interpreted. I do question whether the authors’ conclusion that high accuracy and precision were obtained in the underwater condition, given that only three subjects participated.

Reviewer 2 Report

The article is very well written and all the essential information on the issue under negotiation is included.

Also, modern and impressive equipment was used to collect the data both on land and underwater.

In addition, the methods of data processing and analysis are comprehensive and appropriate.

Nevertheless, there are some weaknesses.

In particular, the introduction does not adequately document the necessity of the research. A comprehensive gait analysis must record data from both the movement of the body parts and the interaction of the walker with the environment, i.e. from the reaction force of the ground on land and, in addition, from the drag and lift forces in water. This comprehensive dataset is essential to meaningfully contribute to the understanding of mobility issues and the potential enhancements achieved through rehabilitation protocols.  It is therefore imperative to use force platforms and kinematic analysis systems, which can also be used to easily identify critical events in the gait cycle. Thus, it would greatly enhance the introduction to incorporate clinical scenarios wherein the methodology elucidated in the article would prove indispensable.

In addition, the novelty of this particular study compared to previous studies is not adequately presented. This omission obscures the scientific advancements achieved by this research in relation to existing knowledge in the field.

Another notable limitation of the study pertains to the selection of participant samples. Specifically, the recruitment of able-bodied individuals without motor impairments appears incongruous, considering that gait analysis primarily serves to diagnose motor issues and gauge the evolution of rehabilitation among individuals primarily affected by neurological and musculoskeletal disorders.Identifying gait cycle events in people with mobility problems is a particular and much greater challenge than in normal walking, as often the initial contact is not made with the heel and it is not always the toes that take off last. Hence, the authors should elucidate the transference of the proposed gait event tracking methodology from healthy individuals to those afflicted with motor deficits, thus ensuring successful implementation in individuals with impaired mobility.

Furthermore, in the methodology it is not clear how many gait cycles were analyzed in each condition. Figure 1 shows 10 gait cycles for walking without weight support on land and 5 for the other conditions. It should also be justified why more gait cycles (e.g. 20) were not analysed. Given that the locomotion occurred on an instrumented treadmill, which inherently provides a larger pool of gait cycles, the rationale behind the chosen quantity should be elucidated.

Regarding the training of the model (as detailed in lines 175-191), it's imperative to specify whether the 80% data allocation for training (consequently, 20% for testing) pertains to the total gait cycles (encompassing cycles per individual across participants) or the number of participants (12 for land and 3 for water). This distinction is crucial to ensure the clarity of the training procedure.

At this point, it's essential to underscore the notable limitation stemming from the small participant cohort, a concern acknowledged by the authors themselves. It's incumbent to provide reasoning for the decision to not include a larger participant group. Especially when considering individuals without health-related issues, the expansion of the participant pool could have been plausible. In particular, a robust justification should be provided for the inclusion of merely 3 participants in the water immersion walking condition.

Finally, with respect to the comparison of angular velocity waveforms between conditions, it's necessary to outline the methodology employed in calculating the correlation coefficient. This detail will provide clarity on the approach used to quantify the similarity of the waveforms.

Reviewer 3 Report

It is quite scarce when such well-written papers are submitted at the first instance. So it took great pleasure for me to read the paper and I could not identify any flaws. Even the ideas that seemed as a problems were really well explained in the limitations. 

While I am not an expert in intelligent algorithms everything was clearly presented and explained throughout the paper in an understandable but also scientific way. 

Thus, I must say congrats to the authors on their great work. 

As a minor comment (as it was not the subject of the paper) how do you see the efficiency of the algorithm in interpreting the gait for people with diffrent walking dissorders or other problems at the level of the lower limb?

Round 2

Reviewer 1 Report

I appreciate the authors' revisions and have no additional comments/recommendations.

Author Response

We appreciate your feedback and decision.

Reviewer 2 Report

I would like to thank the authors for their efforts in addressing the concerns raised in my review and for their thoughtful responses to my comments. I am pleased to note that the majority of the issues have been satisfactorily and adequately addressed.

Nevertheless, I would like to draw attention to a specific concern arising from the authors' response to point 5 of my comments. It pertains to the manner in which the authors combined gait cycles from different individuals and conditions to create the training and testing sets. According to their response, they: “ sampled and shuffled gait cycles among subjects and conditions and designated 80% for training and 20% for testing.

It is worth noting that normal human gait exhibits an exceptionally high degree of repeatability. Consequently, each gait cycle of the same individual typically displays consistent kinematic features, particularly in the sagittal plane, across successive cycles. Even variations in walking conditions, such as changes in speed or body weight support, do not significantly alter this kinematic pattern. The only factors that substantially differentiate gait kinematics are external horizontal resistance forces, as demonstrated in this article's findings for underwater walking.

Therefore, when the authors shuffled the highly repeatable gait cycles from all subjects across different conditions of gait on land to form the training and testing datasets (e.g., using 80% of the data for training and the remaining 20% for testing), they effectively ended up with two datasets that contained overlapping patterns of gait kinematic features. This occurs because the training set likely contains gait cycles from all individuals in the sample across all measured conditions, and the testing set again comprises very similar kinematic patterns in gait cycles from the same individuals under the same conditions.

In contrast, this issue does not arise when selecting data for the training set from all gait cycles across all conditions of a subsample of subjects (e.g., 80% of the subjects, not gait cyclesi.e. 10 out of 12 subjects), with the remaining individuals designated for the testing set (i.e., 2 out of 12 subjects). 

A clear distinction between the training and testing sets is also achieved when the training set includes shuffled gait cycles from all subjects across all land walking conditions, while the testing set consists of shuffled gait cycles from the same subjects but underwater walking conditions. This distinction is evident in the low accuracy (71%) observed when the authors used this dataset configuration.

Similarly, the accuracy in predicting gait events improved (94%) when the authors employed shuffled gait cycles from all individuals in all underwater walking conditions for the training set, with the remaining shuffled gait cycles from all subjects designated as the testing set for underwater walking conditions. Once again, the correct approach for the underwater gait involves selecting a subset of individuals (not gait cycles) for the training set and the remaining individuals for the testing set. However, the limited sample size for underwater walking makes this selection more challenging compared to walking on land.

Based on the above considerations, it is advisable to revisit the algorithm for predicting walking events and configure the training and testing data sets as suggested. Of course, this revision may not be necessary if there are valid arguments to counter the reasoning outlined above.

Author Response

We sincerely value your feedback regarding the data shuffling among subjects and your emphasis on the clear distinction between the training and testing sets. In response to your insights, we have made updates to the methods and results. Specifically, for the BWS (Body Weight Support) data aimed at predicting BWS-related gait events, we have incorporated data exclusively from 10 subjects for training and have utilized data from an additional 2 subjects for model testing. In the context of predicting UW-related gait events, we have taken a similar approach. Our updated model is now trained using data from 2 subjects and tested the model's performance using the remaining dataset from 1 subject.

We made accompanying edits throughout the manuscript to reflect our updated methods and results. Thank you for your input and contributions to our work.

Round 3

Reviewer 2 Report

After thoroughly examining the manuscript in its most recent revised version, I found that all of my feedback has been effectively addressed.

I have no further comments to make.

Congratulations to the authors for their diligent work.